# Health Status of Bycaught Common Eiders (*Somateria mollissima*) from the Western Baltic Sea

**DOI:** 10.3390/ani12152002

**Published:** 2022-08-08

**Authors:** Luca A. Schick, Peter Wohlsein, Silke Rautenschlein, Arne Jung, Joy Ometere Boyi, Gildas Glemarec, Anne-Mette Kroner, Stefanie A. Barth, Ursula Siebert

**Affiliations:** 1Institute for Terrestrial and Aquatic Wildlife Research, University of Veterinary Medicine Hannover, Foundation, Werftstraße 6, 25761 Büsum, Germany; 2Department of Pathology, University of Veterinary Medicine Hannover, Foundation, Bünteweg 17, 30559 Hannover, Germany; 3Clinic for Poultry, University of Veterinary Medicine Hannover, Foundation, Bünteweg 17, 30559 Hannover, Germany; 4National Institute of Aquatic Resources, Technical University of Denmark, 2800 Lyngby, Denmark; 5Friedrich-Loeffler-Institute, Federal Research Institute for Animal Health, Institute of Molecular Pathogenesis, Naumburger Str. 96a, 07743 Jena, Germany

**Keywords:** Wildfowl, sea duck, pathology, parasites, health monitoring

## Abstract

**Simple Summary:**

We performed post-mortem investigations of 121 Common Eiders (*Somateria mollissima*), which were incidentally caught in fishing gear from 2017 to 2019 in Denmark. The aim of the study was to obtain an overview of health issues and pathogens occurring in the population of these birds. The European population of the Common Eider is endangered, but the reasons for the decline of the population have not yet been determined. In times of accelerating species loss, it is important to determine factors that impact population numbers of declining species. The post-mortem investigations included biometric measurements and determination of age, sex and nutritional status, as well as parasitological, bacteriological and virological investigations. The majority of Common Eiders had a good or moderate nutritional status. Most animals were infected with intestinal parasites, and we commonly found inflammation in organs like the liver, kidneys, intestine and the oesophagus. In three animals, a pathogenic bacterium caused inflammatory lesions in several organs. We did not find signs for epizootic diseases or pathologies, which would explain the declining species numbers.

**Abstract:**

The Common Eider (*Somateria mollissima*) inhabits the entire northern hemisphere. In northern Europe, the flyway population reaches from the southern Wadden Sea to the northern Baltic coast. The European population is classified as endangered due to declines in Common Eider numbers across Europe since 1990. In this study, we assessed 121 carcasses of Common Eiders, captured incidentally in gillnets in the Western Baltic between 2017 and 2019. The most common findings were parasitic infections of the intestine by acanthocephalans in 95 animals, which correlated with enteritis in 50% of the cases. Parasites were identified as *Profilicollis botulus* in 25 selected animals. Additionally, oesophageal pustules, erosions, and ulcerations, presumably of traumatic origin, were frequently observed. Nephritis and hepatitis were frequent, but could not be attributed to specific causes. Lung oedema, fractures and subcutaneous haemorrhages likely resulted from entangling and drowning. Two Common Eiders had mycobacterial infections and in one of these, *Mycobacterium avium* subspecies (ssp.) *avium* was identified. This study gives an overview of morphological changes and infectious diseases from one location of the European flyway population. It contributes to future health studies on Common Eiders in the Baltic and Wadden Seas by providing baseline information to compare with other areas or circumstances.

## 1. Introduction

The Common Eider (*Somateria mollissima*) inhabits the entire northern hemisphere, with areas stretching from North America across Europe to Asia [1]. The Baltic/Wadden Sea flyway population stretches from the Dutch Wadden Sea in the South to the entire Baltic area and as far north as Finland and southern Norway [2]. Migration patterns are diverse: while females show high philopatry, males tend to disperse and travel longer distances [3]. Movements within the Baltic Sea and across the borders are common, and migration from hatching sites in the Netherlands to the Baltic area and up to Finland have been recorded [3,4,5,6]. 

In the 20th century, the population size of Common Eiders in Europe was stable and increasing [7,8]; however, since the 1990s, declines have been observed all over Europe due to yet undetermined causes [2,9,10,11]. The first decade of decline was mainly attributed to a drop in wintering populations in Denmark and to a lesser extent in Germany, the Netherlands and Norway [2], but Ekroos et al. [9] described a drastic decrease in breeding numbers from 2000 onwards as well, pointing out that further declines of the flyway population could be expected in the future. Due to these ongoing declines, the European population is now listed as endangered [1]. Many factors may be contributing to the observed decline, including increased predation pressure [12,13,14], inappropriate hunting management [11], reduction in food availability linked to natural yearly fluctuations in mussel stocks, and intensification of commercial mussel exploitation [15,16,17], as well as *Pasteurella (P.) multocida* epidemics [18,19].

Despite these continuous declines, to our knowledge no long-term post-mortem monitoring of the health status and pathology of Common Eiders along the Baltic and North Sea coastline has been conducted so far. However, several pathological studies have been initiated following unusual mortality events, such as an increased death rate observed in a breeding colony of Common Eiders in the Netherlands in the 1980s [18]. A full necropsy of a subset of animals revealed that *P. multocida* was the causative agent. Avian Pasteurellosis, also known as avian cholera or fowl cholera, results from infection by the Gram-negative bacterium *P. multocida*, and can cause lethal infections in wild birds. It is highly contagious, and breeding grounds of Common Eiders, with aggregation of large groups of birds, constitute a perfect environment for the spread of the disease [18,20,21]. Similar outbreaks were observed in Denmark in 1996, when an epizootic caused massive losses of female Common Eiders in breeding colonies, with up to 95% of animals dying from the infection [19], and in 2001, affecting different species of wild and domestic birds [22].

Necropsies of Common Eiders during other mortality events in the Netherlands (1999/2000) and Denmark (2005, 2015 and 2016) were associated with severe emaciation and high parasite loads in the gastrointestinal tract. Food shortage and low food quality were considered the main factor leading to starvation of the birds, although high parasite loads were also noted to play a possible role as an additional stressor [17,23,24].

In the Dutch study of Camphuysen et al. [17], acanthocephalans were identified as *Profilicollis (P.) botulus*, while other studies found *Polymorphus minutus* to be the dominating species [24,25]. Parasitic infestations correlated with enteritis or dilated intestines and occasionally penetrations of the intestinal wall to different degrees [17,24,25,26]. Other parasite species infecting Common Eiders from the North and Baltic Sea are the gizzard worm *Amidostomum (A.) acutum*, intestinal flukes (*Echinostoma* spp.), *Psilotrema* spp., Microphallidae, Hymnolepididae, *Capillaria nyocinarum* and *Cryptocotyle concavum* [17,23,24,27,28,29]. High degrees of parasite infections, especially involving acanthocephalans, have been considered to play an important role in unusual mortality events and epizootics of Common Eiders [30], adding stress on the animals to environmental factors such as low food quality or availability [17,23,24,29]. Experimental infections of Common Eider ducklings have also shown that acanthocephalan infections cause slower weight gain, possibly affecting duckling survival [26]. Thieltges et al. [29] also noted that many studies rely on biased samples of birds that have been washed ashore, and which are therefore more likely to be sick and carry high parasite loads. They investigated parasites in oiled Common Eiders after the wreckage of a freighter and found comparable numbers of parasites in all birds, with no indications of increased mortality in the same season [29].

Viral infections constitute another possible source of disease and increased mortality in the Baltic/Wadden Sea flyway population. In 1996, Hollmén et al. [31] isolated a reovirus of the genus *Orthoreovirus* from the bursa of Fabricius of Common Eider ducklings in Finland, which was potentially linked to increased duckling mortality. Two years later, an increased die-off of males was observed during the mating period in southwest Finland. Affected birds were necropsied and displayed reduced muscle weight, atrophied livers, and necrotic intestinal mucosa. An adenovirus was isolated from cloacal swabs and cloacal tissue and was suspected to play a role in the unusual mortality of the animals [32]. Another study from Hollmén et al. [33] found antibodies for infectious bursal disease virus (IBDV) in nesting Common Eiders. The authors hypothesised that the virus might cause poor fledgling success in Common Eiders if the pathological effect, which is yet unknown, is comparable to that in poultry. Common Eiders are also susceptible to fatal infections with Anatid Herpesvirus-1, the causative agent of duck virus enteritis or duck plague [34]. Furthermore, Common Eiders and waterfowl in general are carriers of avian Influenza virus (AIV), forming a reservoir for the virus and therefore constituting a risk for virus incursion into poultry farms [35]. The seroprevalence in incubating Common Eiders from Danish breeding colonies was between 33 and 71%, although only 12% were positive for the subtypes H5 and/or H7 [36]. During the most recent outbreaks of highly pathogenic avian influenza (HPAI) in Europe in 2016, 2020 and 2021/2022, cases of avian influenza (AI) in Common Eiders caused by the H5 strain were reported [35,37,38,39]. A low number of positive cases in 2020 indicated that Common Eiders are less susceptible to the virus compared to other wild avian species [37], and in the most recent outbreak, Common Eiders were not among the most commonly reported species [35]. 

It has previously been suggested that diseases may play a role in the observed population declines by causing mass mortalities or subclinical effects in the birds [25]. In this study, we describe macroscopic and histological findings, as well as bacteriological, virological and parasitological results from Common Eiders bycaught in the Western Baltic. The aim was to study diseases and their impact on individual animals, e.g., on the nutrition condition, unrelated to observed mortality events. By collecting and analysing these data, we aimed to obtain an overview on the health status of the Common Eider population in the Western Baltic Sea, especially with regard to subclinical effects that may have been overlooked in the context of the declining population. The study delivers a gross overview on morphological changes and infectious diseases and can contribute to other studies monitoring the health of living animals. 

## 2. Materials and Methods

The study material consisted of 121 Common Eiders captured incidentally (bycaught) in gillnet fishing gears in the Sound (Øresund), a strait that connects the Baltic Sea with the Kattegat (Figure 1) between the 23 January 2017 and the 12 March 2019 (Figure 2). We consider the samples investigated in this study to be unbiased, since they were collected regardless of any health issues observed in the population and covered a period of three years and all seasons. All the animals were collected within 24 h after the nets were hauled out, then stored at −20 °C and thawed for approximately 36 h at room temperature prior to dissection.

All 121 animals were examined according to a modified, standardised dissection protocol [40,41,42]. The animals were weighed, and a range of measurements were taken according to Eck et al. [43].

External features including signs of moult, fouling, presence of an incubation patch, state of preservation and external lesions were recorded. The nutritional condition was categorised (mortally emaciated, poor, moderate, and good) based on the condition of the pectoral muscle, the subcutaneous and the intestinal fat, as described by van Franeker [41].

Internal organs, as well as pectoral and leg muscles, subcutaneous fat around the legs and abdominal fat were also weighed. The gonads were weighed and measured and the length and width of the bursa of Fabricius was measured, when present. Based on the developmental stage of the gonads and presence and size of the bursa of Fabricius, the animals were grouped into four age classes (juvenile, immature, subadult or adult) according to van Franeker [41].

### 2.1. Histology

Tissue samples from all organs were collected and fixed in 10% phosphate-buffered formalin, embedded in paraffin wax, cut at 4 µm and stained with haematoxylin and eosin (H&E). Selected tissue sections were stained with Congo red, and tissue sections were subsequently examined under polarised light. In addition, selected tissue sections were stained using Ziehl-Neelsen’s, Grocott’s methenamine silver, von Kossa’s, Sirius red and/or Turnbull’s blue stain. 

### 2.2. Microbiology and Virology

For bacteriological examination, a subset of 28 animals was analysed. The animals were selected based on collection year, age and sex so that all groups were represented. Liver, spleen, kidney, lungs, brain, intestine and, for selected animals, oesophagus were examined by inoculation on Columbia sheep blood (CSB) agar plates and Cystine-Lactose-Electrolyte-Deficient (CLED) agar plates and incubated at 36 °C for 48 h. After production of clean subcultures, isolates were identified by partial sequencing of the 16S-rRNA-gene [44] using BLAST from the US National Library of Medicine.

Based on the results of the pathohistological examination, samples of liver, kidney and spleen of one animal were analysed for the presence of mycobacteria, as described earlier [45]. In brief, approximately 1 g tissue of each organ was homogenised, decontaminated (1% NALC-NaOH) and subsequently cultivated on Coletsos/PACT and Löwenstein-Jensen/PACT agar slants (both Artelt-Enclit, Borna, Germany), as well as in 7H9 broth. The media were incubated at 37 °C and checked every week for bacterial growth. Mycobacterial colonies were identified using end-point polymerase chain reactions (PCRs) targeting 16S rRNA as well as IS*901* and IS*1245* [46,47,48].

For viral examination, samples were stored in plastic bags at −70 °C. Lung, trachea and liver of a subset of 25 animals (same subset as above) were analysed by real time-polymerase chain reaction (RT-PCR) or PCR for the presence of Newcastle Disease Virus (NDV), Avian Influenza virus, and Anatid Herpesvirus 1, respectively. We used the following PCR protocols and kits: for the detection of NDV and avian Paramyxovirus type 1-virus (Orthoavulavirus)-specific RT-PCR we followed a previously published protocol [49] using the SuperScript™ III One-Step RT-PCR kit (Invitrogen/ThermoFisher, Wesel, Germany); for AIV we used the Kylt IVA beta RT-qPCR FLI-C 024 LD (Anicon, Höltinghausen, Germany); and for the Anatid Herpesvirus 1 we used the Kylt^®^ DEV LD (Anicon, Höltinghausen, Germany). RNA and DNA were isolated with the Nucleo-Spin Virus Columns (Machery-Nagel, Düren, Germany).

### 2.3. Parasitology

The prevalence of acanthocephalan parasites was assessed macroscopically and semi-quantitatively categorised as mild, moderate or severe [50,51]. Parasites were collected in tap water and transferred to 70% ethanol after one hour. Other parasitic infections were noted on occasion, when grossly visible or when parasites were visible in the routine histologic examination. Acanthocephalans of a subset of 25 animals were identified further. Selected individuals were identified by PCR as a reference and the remaining specimens were identified based on morphological characteristics, using a stereomicroscope (Olympus SZ61 Stereo Microscope, Olympus, Tokyo, Japan) [52].

### 2.4. Molecular and Phylogenetic Analyses of Acanthocephalans

To achieve molecular parasite identification, genomic DNA was isolated from one acanthocephalan specimen each of ten Common Eiders using the QIAamp Tissue Kit (Qiagen, Hilden, Germany). DNA concentrations were determined using Qubit^®^ 1X dsDNA HS Assay Kit (Invitrogen/ThermoFisher, Wesel, Germany) on a Qubit 4-fluorometer (Invitrogen/ThermoFisher, Wesel, Germany), and quality was determined with a Nanodrop 2000c spectrophotometer (Peqlab Biotechnologie GmbH, Erlangen, Germany). A partial sequence of the mitochondrial gene encoding cytochrome c oxidase subunit 1 (COI) from these acanthocephalan samples was amplified in a PCR using Cox1FW and Cox1Rev primers [53]. Amplification products were Sanger sequenced (Microsynth Seqlab GmbH, Göttingen, Germany), and sequences were compared to those on GenBank using BLASTN. A consensus sequence was submitted to GenBank (Accession number: OP051092). Voucher specimens were deposited at Senckenberg Research Institute and Natural History Museum, Frankfurt/Main (Registration number: SMF 17070, SMF 17071).

For phylogenetic analyses, the obtained consensus sequence was aligned with published Polymorphidae sequences on GenBank using MAFFT [54]. *Gorgorhynchoides bullocki* was designated as an outgroup. GBlocks version 0.91b [55] was used to cut significant gaps in the resulting alignment (default parameters, allowed gap position—“with half”). A maximum likelihood phylogenetic tree with 1000 bootstrap replicates was constructed in MEGA X software [56] using the GTR+I+G model selected based on Akaike Information Criterion (AIC) in jModeltest2 v 0.1.11 [57]. Pairwise genetic distance was determined in MEGA X software [58].

### 2.5. Statistical Analyses

Fisher’s exact test was used to investigate statistical differences (α = 0.05) between biological parameters and the most common pathological findings, i.e., sex, age and nutritional status and parasite infection, enteritis, hepatitis, nephritis and orchitis. Furthermore, the correlation between parasite infection and enteritis was assessed. When a significant association was observed in groups with more than two factor levels (e.g., age group), pairwise Fisher’s exact test was performed as a post hoc test to determine the correlations within the groups. Statistical analyses were conducted in R version 3.6.1 [59], and the ‘rstatix’ package [60] was used for pairwise Fisher’s exact test.

## 3. Results

Approximately two-thirds of the 121 animals were male (*n* = 79) and 42 were female, of which 57 and 21 were adults, respectively. Nine Common Eiders from both sexes were juvenile and four were immature, while nine males and eight females were subadult (Table 1).

Most animals were in a good (*n* = 53) or moderate (*n* = 54) body condition, while only 12 were in poor nutritional condition, and two were mortally emaciated (Table 2). All histopathological findings, listed by organ systems, can be found in Table 3.

Only one adult Common Eider had no pathological findings, it was in a moderate nutritional condition. Six animals (five adults, one subadult) only had parasite infections, and were in a moderate (*n* = 2) or bad (*n* = 4) nutritional condition. Two animals (one adult in a good condition, one subadult in a moderate condition) had parasitic infections and correlating inflammation of the organ. All 112 other Common Eiders had one or more additional pathological findings, in up to 14 different affected organs.

### 3.1. Skin and Subcutis

One of the major findings in all animals was subcutaneous haemorrhage, which was seen along the neck, thorax, abdomen and inner thighs in 66 (54.5%) animals. Two animals had mild abrasions of the skin on the legs. A mild, pyogranulomatous panniculitis was diagnosed histologically in one case. Two Common Eiders had foreign bodies in their subcutaneous tissue, which were identified as pellets of lead and iron shots (Figure 3).

Neither of the two animals showed any grossly visible inflammatory reactions in the surrounding tissues. One of the pellets left an imprint on the surface of the liver, but did not cause any histologically visible signs of reacting tissue, either. 

### 3.2. Bones and Beak

Fractures of the beak and the bones of the extremities were recorded frequently. In nine cases, the beak was fractured, and in 21 cases, different parts of the legs were affected.

### 3.3. Gastrointestinal Tract

In almost 75% of the study sample (*n* = 89 animals), focal/multifocal mucosal alterations of the oesophagus were detected (Figure 4 and Figure 5), characterised by white-yellowish elevated plaques of varying number and size. For 82 of these animals, histological examination revealed partially overlapping inflammations with erosive (*n* = 1), ulcerative (*n* = 61), necrotising (*n* = 33), pustular (*n* = 36) and granulomatous (*n* = 2) character. Additional findings included epithelial hyperplasia (*n* = 4) and hyperkeratosis (*n* = 2). For the remaining seven Common Eiders, macroscopically visible lesions resembled those of the other animals, but due to the size and focal distribution of the lesions, no histological characterisation was possible. 

The proventriculus and gizzard were altered significantly less often than the upper digestive tract. Focal (*n* = 9) and diffuse (*n* = 1) gastritis was diagnosed in ten (8%) Common Eiders, affecting the proventriculus in three, the gizzard in six, and both stomachs in only one case. Histologically, the inflammations were characterised as lymphocytic/follicular (*n* = 4), eosinophilic (*n* = 3), ulcerative (*n* = 2), necro-suppurative (*n* = 2), granulomatous (*n* = 1) and/or fibrotic (*n* = 1), in some cases occurring simultaneously. In one case, the granulomatous and necro-suppurative gastritis of the gizzard correlated with a granulomatous serositis and was grossly visible as multiple irregular granulomas on the mucosa, up to 2.5 cm in diameter, partially perforating the gastric wall (Figure 6).

Two individuals had sharp pieces of mussels perforating the cuticula and penetrating the mucosa underneath (Figure 7). Accumulation of amyloid in the proventriculus was diagnosed twice, and one animal had a fibrosis of the gizzard. Amyloidosis occurred in the proventriculus, spleen, kidneys, thymus, liver, thyroids, adrenal glands, intestine and/or testes of four animals as described in the following sections.

Enteritis was seen in 43 (35.5%) Common Eiders. For 26 animals, inflammatory changes were seen histologically and specified further as partially overlapping granulomatous-pyogranulomatous (*n* = 15), lympho-plasmacytic (*n* = 10), ulcerative (*n* = 3) and necrotising (*n* = 1). In the remaining 17 cases, a focal granulomatous/pyogranulomatous enteritis was diagnosed based on the morphology of the macroscopically visible alterations and comparison with the aforementioned histological results (Figure 8), but due to the size and focal distribution of the lesions, no histological characterisation was possible. Table 4 shows the interrelation between enteritis and the nutritional condition of the animals. There was no statistically significant correlation between enteritis and the nutritional condition (*p* = 0.64) or sex of the animal (*p* = 0.08). However, enteritis correlated with the age of the animals (*p* = 0.002). When pairwise Fisher’s test was applied, a statistically significant difference was observed between adult and immature animals, with an increased probability of immature animals having enteritis (*p* = 0.006). In two cases, the enteritis was associated with a serositis of the intestine, classified as granulomatous or fibrino-suppurative. Fibrosis was correlated with enteritis in three animals and diagnosed alone once. Amyloidosis in the wall of the small intestine was seen in one animal.

In a subset of 18 (14.9%) Common Eiders, staining with Sirius red revealed mild to moderate eosinophilic infiltration of the mucosa independently of an intestinal parasite infestation.

### 3.4. Liver

Sixty-five (53.7%) Common Eiders were diagnosed with hepatitis histologically, which was specified as focal or multifocal and mild to moderate lymphohistiocytic hepatitis in most cases (*n* = 64). In one case, a mild purulent hepatitis was additionally diagnosed. One Common Eider had a severe necro-suppurative to pyogranulomatous hepatitis with acid-fast, rod-shaped bacteria, indicating a mycobacterial infection. 

No statistically significant correlation was observed between the occurrence of hepatitis and the age (*p* = 0.41) or nutritional condition (*p* = 0.25) of the animal. However, the occurrence of hepatitis seems to be influenced by the sex, with a statistically significant difference between male and female Common Eiders (*p* = 0.007), indicating that females are more often affected. 

Focal fibrosis was seen in three animals, amyloidosis in two, and proliferation of bile ducts once. 

### 3.5. Urinary and Reproductive Tract

In 50 (41.3%) animals, the kidneys appeared congested macroscopically. Interstitial nephritis and/or pyelitis was diagnosed histologically in 69 (57%) Common Eiders and specified as lymphohistiocytic and plasmacytic in most cases (*n* = 66). In one case, the inflammation was necrotising-granulomatous, one Common Eider had a necro-suppurative nephritis with abscess formation, and in one case, a histological identification was not possible due to the poor preservation status of the tissue. 

No statistically significant correlation was observed between the occurrence of nephritis and the age (*p* = 0.44) or nutritional condition (*p* = 0.62) of the animal. However, the occurrence of hepatitis seems to be influenced by the sex, with a statistically significant difference between male and female Common Eiders (*p* = 0.001), indicating that females are more often affected.

Multifocal amyloid deposition was seen three times affecting glomerular mesangial and medullary interstitium. 

Alterations of the female reproductive organs were noted twice, specifically as lympho-histiocytic oophoritis and mural hyalinosis of ovarian vessels. Pathological findings of the testes were more common, with focal or multifocal orchitis diagnosed in 32 (40%) male Common Eiders. It affected adult (*n* = 22) and subadult (*n* = 4) as well as juvenile (*n* = 3) and immature (*n* = 1) animalsm and no statistically significant effect of age was observed (*p* = 1). For most of the affected individuals, the inflammation was lymphocytic (*n* = 29); more rarely, it was purulent (*n* = 1), necrotising (*n* = 1) or granulomatous (*n* = 1).

Independent from the orchitis, no signs for spermatogenesis were observed in 56 (70.9%) male Common Eiders. In comparison, follicles of different sizes and in different phases were present in 37 (88%) female Common Eiders. 

### 3.6. Haematopoietic and Endocrine System

Splenitis was detected in three Common Eiders and characterised as granulomatous and necrotising. In one case, acid-fast, rod-shaped bacteria were seen histologically, indicating a mycobacterial infection. Amyloidosis was observed in three individuals in the spleen as well as in the adrenal glands, while it was only diagnosed twice in the thyroids. 

In adult Common Eiders, variable amounts of brownish pigment were observed in the cortical cells of the adrenal glands (*n* = 18, 14.8%), most likely representing age-related lipofuscin. 

### 3.7. Cardiovascular System and Lung

In 118 (97.5%) cases, the Common Eiders had oedematous lungs, and in 121 (100%) animals, hyperaemia of the lung was detected. Only one bird had a pulmonary emphysema, and another one moderate haemorrhages in the pulmonary tissue. A lymphocytic aerosacculitis was diagnosed once. 

The only pathological finding of the cardiovascular system was a moderate fibrosis of the atrioventricular valve in one adult female. 

### 3.8. Parasitic Infections

By far the most commonly affected organ was the intestine (Figure 8). In 95 (78.5%) cases, a parasitic infection with acanthocephala was recorded during necropsy. In 20 of these animals, different developmental stages of parasites were also detected via histological examination, including eggs in four animals and adult specimens of metazoan parasites, which were not identified further. In three additional cases, metazoan parasites were only visible histologically. In approximately 45% of all cases (*n* = 42), the acanthocephalans were correlated with enteritis, which was predominantly granulomatous/pyogranulomatous (*n* = 37), or in fewer cases, lymphocytic (*n* = 10), ulcerative (*n* = 3) and necrotising (*n* = 1).

Parasite identification at species level was performed for 25 animals and *Profilicollis botulus* was identified in all cases. This was confirmed by molecular identification. The ten sequences obtained in this study were identical (pairwise distance between 0.000 and 0.011). BLASTN results showed a 100% and 99% identity to *Profilicollis botulus* isolated from Wadden Sea crabs (Accession no: KX279920) and Common Eiders in Denmark (Accession no: EF467862), respectively. Phylogenetic analyses further strengthened this classification, as the sequence was placed within a monophyletic *Profilicollis* clade (Appendix A).

For nine animals, the aforementioned staining with Sirius red showed an infiltration of the mucosa with eosinophilic granulocytes. However, such infiltration was also visible in nine Common Eiders in which no parasites were detected. 

Table 5 shows the interrelation between parasite burden and the nutritional status of the animals. The occurrence of acanthocephalan parasites did not show a statistically significant correlation with age (*p* = 0.71), sex (*p* = 0.24) or the nutritional condition (0.64) of the animals. However, a statistically significant correlation was observed between the presence of parasites and the occurrence of an enteritis (*p* = 0.0001).

Parasite eggs were found in the bile ducts inside the liver of one animal. Histological examinations also revealed kidney infection in four individuals, which were suspected to be protozoans. In six animals, nematodes were seen histologically in or beneath the koilin layer of the gizzard and were associated with gastritis in only two cases. One of the Common Eiders had a foreign body penetrating the gastric mucosa. 

### 3.9. Virology

Of the 25 examined animals, none was positive for Newcastle Disease Virus, Avian Influenza virus or Anatid Herpesvirus 1.

### 3.10. Bacteriology

A detailed overview of all bacterial species that were detected in the different organ systems is shown in Table 6. Bacterial growth was mild (≤10 colonies/petri dish) for all listed isolates, except *Mycobacterium avium* subspecies (ssp.) *avium*. For the 28 animals that were part of the microbiological screening, no pathogenic bacteria were detected. 

Histologically, one adult female Common Eider had a chronic, granulomatous splenitis, and acid-fast, rod-shaped bacteria were detected in the tissue. Another adult male animal had acute, necrotising splenitis and acute, purulent necrotising hepatitis and nephritis with acid-fast, rod-shaped bacteria in the liver tissue. These findings indicated a mycobacterial infection, and analysis of the liver, kidney and spleen samples of the latter indicated results that were positive for *Mycobacterium avium* ssp. *avium* infection, confirming the presumption.

## 4. Discussion

The present dataset of 121 Common Eiders provides the first overview of health data over a multi-year period and reflects the health of the population independent of a precise mortality event. Apart from one animal, all animals had one or multiple pathological findings, with intestinal parasite infection, hepatitis, nephritis, and oesophagitis being the most frequent findings. Bacterial infections with *Mycobacterium avium* were detected in three cases, and in all other animals, the changes could not be linked to any infectious agents. 

The continuous decline of the Common Eider population in the Baltic/Wadden Sea flyway since the 1990s has raised many questions about factors influencing population dynamics and the underlying reasons for this decline. Despite different health issues such as high parasite loads, emerging viral diseases, and bacterial infections being discussed as potential factors, to the best of our knowledge, no long-term pathological monitoring or continuous post-mortem studies have been published in Europe to date.

As the Common Eiders in this study were confirmed bycatches, and all animals were directly handed over by the fishermen, we concluded that several pathological findings can be attributed to the process of struggling in the net and drowning. Most striking were the lung oedema and hyperaemia, observed in nearly all animals. Lung oedema in birds does not present as prominently as in mammals, with froth in trachea and the upper airways [61]; nevertheless, frothy liquid was visible in the thorax and on the cut surfaces of the lung and oedema was confirmed histologically. Despite the congestion of the lungs in all 121 animals, other organs did not commonly show signs of congestion. Fluid was observed regularly in kidneys and liver, when the organs were cut, but it is assumed that the fluid was probably associated with the freezing and thawing of the animals. Saturated plumage was observed but not assessed or noted systematically nor considered as a valuable characteristic for drowning, as all animals were frozen prior to examination. 

The fractures of the bones and/or beak, as well as abrasions (*n* = 2) and subcutaneous haemorrhages (*n* = 66), can be attributed to entanglement in the net. 

Bycatch is considered a major threat to several seabird species, including the Common Eider, with thousands of Common Eiders drowning in gillnets every year in the North and Baltic Sea area [62]. Still, bycatch levels may be difficult to evaluate, as these estimates rely on appropriate monitoring and accurate reporting from fisheries, which are often lacking [63]. Several studies have investigated morphological changes of bycatch in birds, and although no pathognomonic alterations for bycatch exist [64], several findings are typical, and help to distinguish between different causes of death. These include skeletal fractures, abrasion, saturated plumage, subcutaneous haematoma/haemorrhages, lung oedema and multi-organ congestion [61,65,66,67], which is in line with our findings.

Many Common Eiders were infected with acanthocephalan parasites in the small intestine. In 25 cases, the parasites were identified as *Profilicollis botulus*, which is consistent with the findings of previous studies in the Baltic/Wadden Sea area [17,23,27,29] and other geographical locations [68,69]. The low genetic divergence of *P. botulus* across geographic locations and hosts is in agreement with a previous study that found that *P. botulus* sequences were all grouped together in one haplotype network [70]. Molecular tools are invaluable for identifying closely related parasites when morphological characteristics are not sufficient [53]. Acanthocephalans have been linked to severe emaciation and unusual mortality events but the majority of studies on parasitic infections in Common Eiders, even when linked to unusual mortality, strongly suggests that parasites cannot be considered as the sole cause of death [17,23,27,29]. Experiments in Common Eider ducklings showed a slower weight gain when ducklings were infected with parasites [26], indicating that the infection makes the animals more vulnerable to fluctuations in food availability and prone to secondary infectious diseases. Only a few studies mention pathological findings such as enlarged intestines linked to the infection [25,26]. Our results indicate that enteritis is intimately linked to parasite infection with approximately 45% of all infected birds displaying inflammatory changes of the intestine and a statistically significant correlation between enteritis and parasite infection. Granulomatous enteritis was most frequently observed, and has been linked to parasite infection in mammals [71] and birds [72]. Additionally, eosinophilic infiltration of the intestinal mucosa was observed in nine animals. To verify this finding, the intestine of nine selected animals without parasite infection was also stained with Sirius red and examined for the presence of eosinophilic granulocytes. Eosinophilic infiltration has been linked to parasite infestation in mammals and birds [71,72]. However, eosinophils may be present in the healthy intestinal mucosa [73,74,75], and their occurrence has to be interpreted carefully when assessing inflammatory reactions. A marked increase, indicative of an inflammatory response [75], was not observed in the examined Common Eiders, and we therefore consider the eosinophils to be a normal infiltration of the mucosa.

Even though parasites seem to induce enteritis, the parasite burden did not seem to be directly correlated with nutritional status. Furthermore, no obvious link between the grade of enteritis and the nutritional status existed. These observations were confirmed by statistical analysis and provide further evidence that intestinal parasites do not inevitably cause significant health issues or severe weight loss in the Common Eiders but may constitute an additional stressor for the birds in seasons with low food availability, as previously suggested [23,27]. Whether high parasite burden or a bad nutritional condition favours bycatch in seasons with low food availability has to be considered and examined as well. Parasite burden was only measured semi-quantitatively in the current study; therefore, no direct comparison with other studies was possible. It cannot be excluded, however, that the higher loads of acanthocephalan infections that have been observed elsewhere could potentially lead to negative health effects for the affected birds. As our study is limited to parasite assessment and identification in the intestine, i.e., acanthocephalans, no precise statement about the prevalence of other parasite species and their possible health effects can be made. However, parasites in different organs were detected occasionally upon histological examination, indicating that the parasite burden may be underestimated. Borgsteede [28] found high numbers of the gizzard worm *Amidostomum acutum* in dead Common Eiders from the Wadden Sea and discussed their role in mass mortality events of Common Eiders. In the present study, six animals were infected with nematodes in the gizzard. These were detected by histological examination, although species identification was not achieved. In two cases, gastric nematodes were correlated with gastritis, but due to the limitations of our study, we suggest further research to evaluate the prevalence and role of *A. acutum* in Common Eiders. 

The cause of hepatitis, which occurred in a striking number of Common Eiders, was not identified. For 19 animals, a virological and microbiological examination of liver tissue was conducted, but the results did not reveal any infectious agents that could possibly have caused the mild to moderate lymphohistiocytic inflammation. However, we cannot rule out that the lesions were further progressed at the time of examination, and that bacteria were simply no longer present. A study by Garbus et al. [76] found elevated liver enzymes in Common Eiders investigated from a hunting bag, pointing towards possible hepatic disease, but did not perform histology to confirm this presumption. They also found grossly visible liver granulomas caused by non-lethal wounding with steel pellets in two animals. The authors emphasised the need to better understand the risk of non-lethal injuries caused by gunshots. In the present study, no inflammatory reactions were observed in relation to gunshot pellets, which were found in the subcutaneous tissue of two Common Eiders and left an imprint in the liver tissue of one bird. We suggest that more research on the occurrence and significance of liver diseases is conducted in the future, as our findings demonstrate that subtle liver disease is frequent in Common Eiders from the Baltic/Wadden Sea flyway population. Future studies should also seek to determine the possible causes of these underlying hepatic diseases, and look further into the observed differences between male and female Common Eiders. If future studies can confirm a higher susceptibility of female animals to certain diseases, sex-related factors should be considered and examined in more detail. 

Oesophageal lesions were detected macroscopically in 89 birds. All Common Eiders had filled gastrointestinal tracts, so we concluded that the lesions did not significantly impair food uptake. Whether the aetiology involves infectious agents remains unclear, as bacteriological examination was only conducted for the oesophageal tissue of seven animals. Although no bacterial growth was detected in any of these cases, the low sample size does not allow a final statement to be made regarding bacterial involvement. Another possible explanation might be mechanical irritation of the mucosa by mussel shells. As blue mussels constitute one of the main food sources of the Common Eider [16], this would imply that similar lesions occur regularly, and may heal without adverse effects on the wellbeing. The fact that no chronic stages of inflammatory processes were visible also indicates that the lesions heal rather quickly and do not cause major health problems for the animals. Hollmén et al. [32] found necrotic areas in the oesophageal mucosa of two male Common Eiders in association with adenovirus infection. The virological examination in the present study did not include adenoviruses; however, we consider it unlikely that the high prevalence of lesions was of viral origin, as the Common Eiders investigated by Hollmén et al. [32] died during a period of unusually high male mortality and there were other significant pathological findings. Furthermore, no intranuclear adenoviral inclusion bodies were seen in any of the animals.

To investigate whether these oesophageal lesions occur on a regular basis and are corelated with available food sources, i.e., sharp-edged mussels, or are due to another as-yet-undetermined reason, further pathological studies should be conducted. 

Histopathology also revealed a high prevalence of orchitis, mainly affecting adult Common Eiders. With regard to the decreasing population numbers, this deserves special attention, as orchitis may cause infertility, therefore directly affecting reproductive success. In domesticated poultry, orchitis has been associated with bacterial infections of *Escherichia coli* and *Staphylococcus aureus* [77,78], as well as infections of viral origin [79]. The reproductive tissue of seven Common Eiders with orchitis was analysed microbiologically, but no infectious bacteria or fungi were detected. For the same seven animals, PCR analyses for AIV, NDV or Herpesvirus were negative. The non-purulent character of the inflammation did not indicate bacterial involvement. However, due to the limited sample size of bacteriological and virological investigations, the underlying cause of orchitis remains unclear. As the orchitis also affects adult animals, the possibility of it resulting in infertility would be of great concern, as sexually mature animals would be incapable of reproducing. This is reinforced by the fact that 70% of the male Common Eiders did not show signs of spermatogenesis, regardless of the season or age. Based on the mild severity, we would assume that the inflammation does not have a significant impact on reproductive potential. As a lack of spermatogenesis was also observed in otherwise healthy reproductive organs, other factors, such as reproductive toxicity of marine pollutants [80,81,82], need to be considered and assessed further. Studies on the population dynamics of the Common Eider in the Baltic have discussed factors affecting the reproductive success in breeding colonies and concluded that poor food quality, nutrient reduction and foraging conditions, resulting in poor pre-breeding condition of the females may be one of the main drivers [15,83]. However, diseases of the reproductive tract have not been considered in the existing literature, and our results indicate that it would be worth conducting further research to determine the causes and effects on the reproductive capability in male Common Eiders. The brownish pigment found in the adrenal glands of adult Common Eiders (*n* = 18) was interpreted as lipofuscin, and most likely represents an age-related change without clinical relevance.

The virological examinations all turned out negative. A limitation of the study was that no serum samples could be analysed for antibodies due to the freezing and advanced decomposition state of the animals. Although the findings indicate that none of the animals was acutely infected with the examined viruses, we cannot rule out that they are circulating in the population. During recent and ongoing outbreaks of HPAI in Europe, Common Eiders have also tested positive [35,37]. Reports of the distribution and prevalence across Europe indicate an increased species range among wild birds [35] and found high prevalence in some species is not necessarily correlated with high mortality [84]. Therefore, future studies should aim to sample fresh animals and include serological examinations to investigate the prevalence of viral diseases like HPAI in the Common Eider population.

The results of our study indicate that regular monitoring of diseases in Common Eiders can reveal a variety of health issues, which might require further research efforts in order to assess the impact at the population level. Pathological findings like hepatitis and nephritis, which were mild in most cases, might not have a direct effect on the health of the affected individuals, but their high prevalence in our study sample is concerning.

Our findings alone cannot explain the current overall decline in the number of Common Eiders in the Baltic/Wadden Sea flyway population, but they may help understand some of the underlying reasons for this decline. Still, it can certainly be concluded that the variety of pathological findings affects the health of Common Eiders and acts as an additional stressor to the population, especially when food availability and/or quality is low and when environmental and seasonal conditions require resistance, good body condition, and health. The large distribution of the Baltic/Wadden Sea flyway population and migratory movements within this area pose a high threat of disease transmission between different colonies. With respect to the continuous decline of the Common Eider population, the implementation of continuous post-mortem investigation for diseases and morphological changes might provide valuable information about possible factors adversely affecting the health and wellbeing of the population.

As Thieltges et al. [29] proposed, recent studies on pathology and parasite infections in Common Eiders have often consisted of biased samples of dead animals or were collected during unusual mortality events, thus making them non-representative of the health situation of the entire population. We consider the samples investigated in this study to be unbiased, since they were collected as bycatch from fishermen regardless of any health issues observed in the population and covered a period of three years and all seasons. Bycatch, as long as not fully preventable, can therefore constitute a valuable source for the collection of animals in order to conduct ongoing pathological studies in order to investigate diseases and threats in the Common Eider population. Additionally, dead Common Eiders from hunting bags could be retrieved to collect study material, as was done in other studies in Denmark and Finland [76,85,86]. However, this would restrict sampling sites to countries that allow hunting of Common Eiders, and collection would only cover the hunting seasons and certain age classes and sexes that are allowed to be shot. Regardless of the collection method, continuous health monitoring would allow better understanding of disease dynamics and effects on the individual and the population. 

Even though our study provides a comparatively large dataset over several years, the significance at the population level needs to be evaluated under certain other restrictions. The sample size is still relatively low, especially when it comes to the number of bacteriological and virological samples. Another limitation is the confined sampling area. Especially in a highly mobile species like the Common Eider, more sites should be investigated to reflect the whole species range. As samples were collected over several years and frozen prior to investigation, certain findings may have been disguised by the process of freezing and thawing. Especially for bacteriological analyses, but also with the addition of further examinations like blood parameters, it would contribute to our findings to analyse better-preserved animals. Last but not least, necropsies were performed by different veterinarians, so perception bias has to be taken into account to some extent, which we tried to minimise as much as possible by using a standardised protocol and assessment guides. 

## 5. Conclusions

This study offers an insight into the prevalence of diseases and pathology occurring in Common Eiders in the Danish Sound and provides information about health issues and stressors potentially affecting these individuals, as well as the entire Common Eider Baltic/Wadden Sea flyway population. It provides a first dataset of pathological findings for this species and indicates that subclinical diseases are very present in the population and may be stressors that contribute to the observed population declines. 

To better understand the disease dynamics and health issues of the population, a greater number of detailed post-mortem investigations should be conducted. Future studies should extend the screening for bacterial and viral agents, which could shed light on disease aetiology and the infectious diseases potentially circulating in the population. Furthermore, analyses of toxic pollutants should be conducted in order to assess other factors possibly contributing to the health of the species. 

To implement specific management actions, it is necessary to determine the underlying causes of the pathological findings. However, any stressors that may contribute to an impaired immune system or poor health of the animal should be included in management plans, e.g., control of invasive predatory species, monitoring of mussel fisheries to ensure food availability, and provision of optimal breeding and feeding environments. Reducing environmental stressors to a minimum will help to avoid the findings causing more serious health problems to individuals. 

## Figures and Tables

**Figure 1 animals-12-02002-f001:**
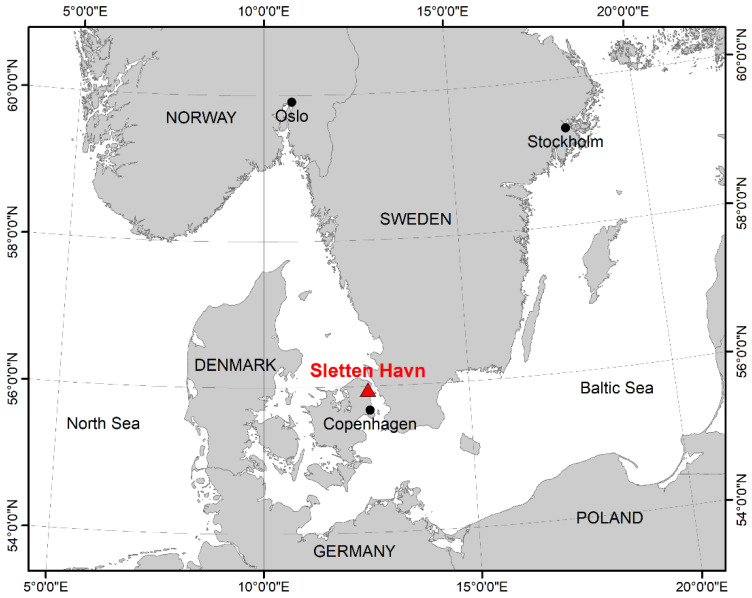
Map of the location of Sletten Haven, Denmark, where all 121 animals were collected.

**Figure 2 animals-12-02002-f002:**
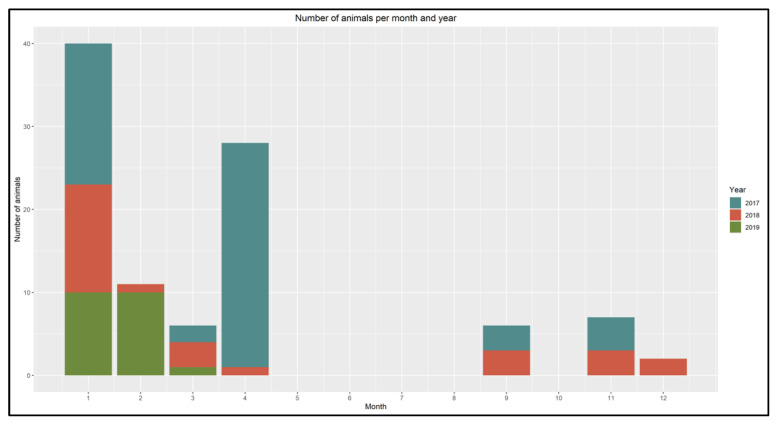
Distribution of bycaught Common Eiders by month and year from January 2017 to March 2019.

**Figure 3 animals-12-02002-f003:**
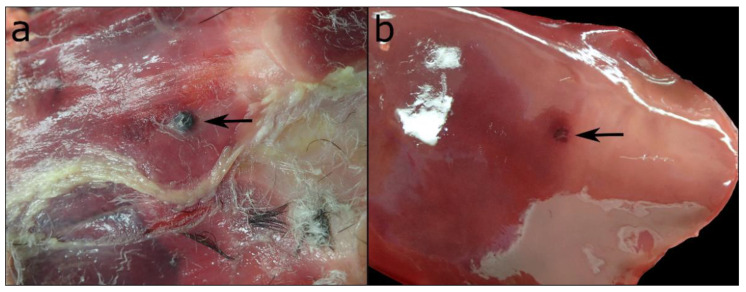
(**a**) Gunshot pellet in the subcutis of the abdomen with (**b**) a focal, superficial impression on the liver tissue of the same bird.

**Figure 4 animals-12-02002-f004:**
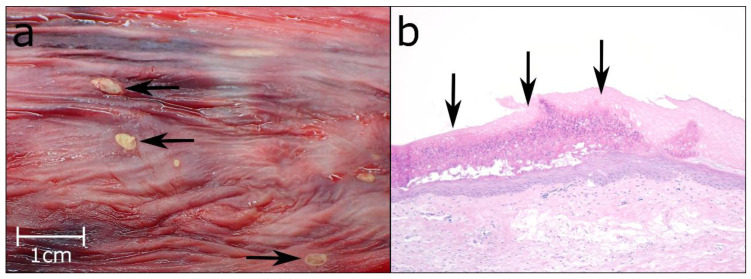
(**a**) Multifocal ulcerations of the oesophageal mucosa (black arrows). (**b**) Pustular lesions in the oesophageal mucosa (black arrows), Hematoxylin & Eosin (HE), 10×.

**Figure 5 animals-12-02002-f005:**
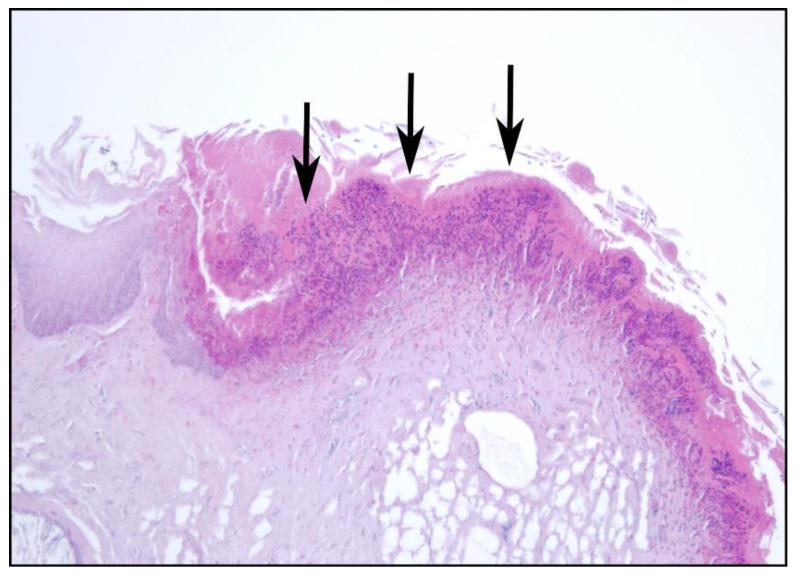
Ulcerative and purulent oesophagitis (black arrows), HE, 20×.

**Figure 6 animals-12-02002-f006:**
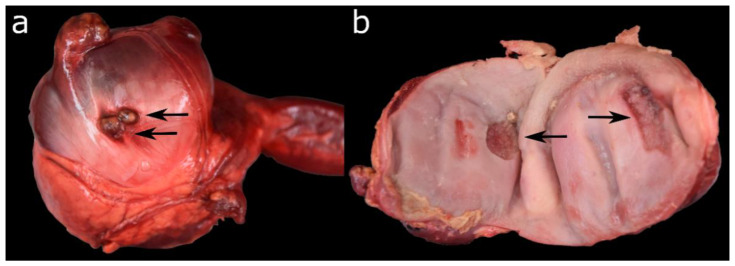
(**a**) Granulomatous and pyogranulomatous serositis of the gizzard. (**b**) Necrotising gastritis of the gizzard of the same animal.

**Figure 7 animals-12-02002-f007:**
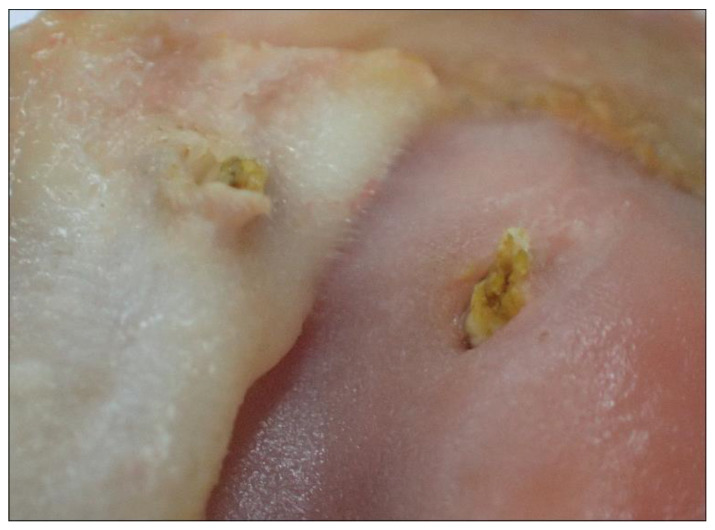
Ulcerative gastritis of the gizzard of a male Common Eider.

**Figure 8 animals-12-02002-f008:**
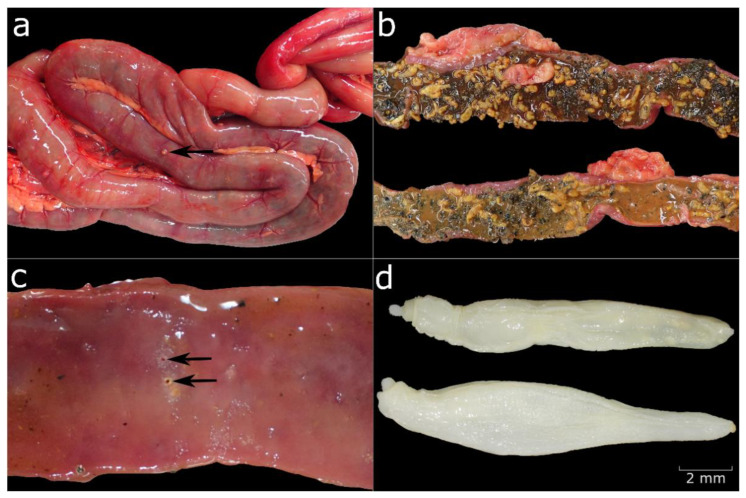
(**a**) Parasite infection of the small intestine with granulomatous enteritis visible on the serosa (black arrow). (**b**) Bright orange parasites in the intestine. (**c**) Ulcerative lesions in the mucosa (black arrows). (**d**) *Profilicollis botulus* under the stereomicroscope.

**Table 1 animals-12-02002-t001:** Age distribution of male and female Common Eiders.

Sex	Age				
	Juvenile	Immature	Subadult	Adult	Total
Male	9	4	9	57	79
Female	9	4	8	21	42
Total	18	8	17	78	121

**Table 2 animals-12-02002-t002:** Nutritional status of 121 Common Eiders in relation to their age.

Age	Nutritional Condition				
	Good	Moderate	Poor	Emaciated	Total
Juvenile	2	5	1	0	8
Immature	5	7	6	0	18
Subadult	7	10	0	0	17
Adult	39	32	5	2	78
Total	53	54	12	2	121

**Table 3 animals-12-02002-t003:** Pathomorphological findings of 121 Common Eiders listed by organ systems.

Organ	Pathological Finding	Number of animals Affected
Cardiovascular system and lung		
Air sacs	Aerosacculitis	1
Heart	Valvular fibrosis	1
Lung	Emphysema	1
	Haemorrhage	1
	Hyperaemia	121
	Oedema	118
Gastrointestinal tract		
Oesophagus	Epithelial hyperplasia	4
	Hyperkeratosis	2
	Oesophagitis	82
Proventriculus	Amyloid deposition	2
	Gastritis	4
Gizzard	Fibrosis	1
	Gastritis	7
	Parasite infection	6
	Cuticular perforation	2
	Serositis	1
Intestine	Amyloid deposition	1
	Enteritis	24
	Fibrosis	4
	Parasite infection	95
	Serositis	2
Liver	Amyloid deposition	3
	Congestion	2
	Fibrosis	3
	Hepatitis	65
	Parasite infection	1
	Proliferation of bile ducts	1
Haematopoietic and endocrine system		
Adrenal glands	Amyloid deposition	3
Spleen	Amyloid deposition	4
	Follicular hyperplasia	2
	Splenitis	3
Thyroid gland	Amyloidosis	2
Skin and bones		
Bones	Fracture	21
Beak	Fracture	9
Skin	Skin abrasion	2
Subcutis	Foreign body	2
	Haemorrhages	66
	Panniculitis	1
Urinary and reproductive tract		
Kidneys	Amyloid deposition	4
	Concrements	1
	Hyalinosis of glomerular mesangium and medullary interstitium	1
	Nephritis/Pyelitis	78
	Parasite infection	4
Ovary	Hyalinosis of ovarian vessels	1
	Oophoritis	1
Testes	Amyloid deposition	1
	Orchitis	32

**Table 4 animals-12-02002-t004:** Semiquantitative assessment of the grade of enteritis in relation to the nutritional condition.

Nutritional Condition	Grade of Enteritis			
	Mild	Moderate	Severe	Total
Good	8	14	3	25
Moderate	8	10	1	19
Poor	1	0	1	2
Emaciated	1	0	0	1
Total	18	24	5	47

**Table 5 animals-12-02002-t005:** Semiquantitative level of acanthocephalan infection in the intestine in relation to the nutritional condition of the animal.

Nutritional Condition	Parasite Infection				
	Mild	Moderate	Severe	None	Total
Good	23	18	4	8	53
Moderate	15	22	4	13	54
Poor	5	3	1	3	12
Emaciated	2	0	0	0	2
Total	45	43	9	24	121

**Table 6 animals-12-02002-t006:** Bacterial species detected in the organs of 29 animals, listed by organ systems.

	Organ								
	Lung	Liver	Intestine	Spleen	Kidneys	Brain	Oesophagus	Reproductive tract	Stomach
Number of Organs Examined	25	26	25	25	26	25	10	22	1
**Bacterial species**									
*Actinomyces marimammalium*	1								
*Bacillus licheniformis*							3		
*Bacillus* sp.		1	2						
*Cardiobacterium* sp.	1								
*Citrobacter* sp.	1								
*Deinococcus* sp.					1				
*Enterococcus faecalis*	1								
*Enterococcus gallinarum*							1		
*Escherichia coli*	1								
*Kocuria* sp.		1							
*Lelliottia amnigena*					1				
*Leucobacter* sp.	1								
*Mycobacterium avium* subspecies *avium*		1		1	1				
*Pasteurellaceae*	1								
*Psychrobacter arenosus*					1				
*Psychrobacter* sp.	1	1			2	1			
*Serratia* sp.						1			
*Staphylococcus hominis*		1							
*Stenotrophomonas rhizophila*		1							
*Stenotrophomonas* sp.	2				1				
*Streptococcus pharyngis*							1		
Total	10	6	2	1	7	2	5	0	0

## Data Availability

All data generated or analysed during this study are included in this published article (and its Appendix A).

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
