# Peer review of "Health Status of Bycaught Common Eiders (Somateria mollissima) from the Western Baltic Sea"

_animals, 2022, doi:10.3390/ani12152002_

Round 1
Reviewer 1 Report
A paper presented by Schick et al. presents pathological and parasitological findings in bycaught common eiders (Somateria mollissima) from the Western Baltic Sea. In this study, authors assessed 121 carcasses of common eiders, captured incidentally in gillnets in the Western Baltic. The manuscript gives an insight into the prevalence of diseases and pathology occurring in common eiders in the Danish Sound and provides information about health issues and stressors possibly affecting the individuals but also the entire common eider Baltic/Wadden Sea flyway population.
Given this report, and the literature, I believe this to be a well-written manuscript- I read it with great interest. The article should find interest among researchers studying the wild life diseases, wildlife conservation, microbiology, histopathology, parasitology and epidemiology. The manuscript is very clear and based on my veterinary experience in molecular biology, pathology and wildlife diseases, the methodology is proper.
The manuscript could be published as it is, however I have some suggestions, that may make it (even) better:
Major flaws:
- The introduction is definitely too long, sometimes even off-topic, so I suggest shortening this chapter to the most necessary information. Some of this information should be transferred to the discussion.
-In general, some information should be transferred between chapters. Please follow the author's guidelines to make the article more readable.
- Please refer to specific legal documents that allowed for these studies to be carried out without the approval of the local ethics committee or special permission. You can't just say that since the animals were dead, no consent was needed
- The biggest flaw of this article is any lack of statistical research. The obtained data are extremely interesting from the point of view of the population of these birds. The authors should show statistical dependencies (or lack thereof) between pathological changes, the presence of parasites or even the years in which the birds were obtained. With this amount of data, the possibilities are many. I suggest using multiple regression, the Fisher test, the Pearson correlation, or multiple regression. Authors could use generalized linear models to evaluate individual years. The aforementioned statistics would significantly enrich the article. I believe that it is necessary for the publication of the article
Minor flaws:
I made a small number of suggestions on the attached file
Conclusion
To sum up, I believe that the article represents a well-done study. Results are important for this endangered species. There is also no such extensive research in the literature. The article has very few things that I could find fault with. Unfortunately, but I think that it is obligatory in this article to conduct statistical research so that the article better presents individual, population or time-environmental relationships. Despite these shortcomings, the entire study has been described very well and the presentation is at a high level. I am pleased to present my opinion to the editors that the article should be accepted after minor revision.
With kind regards

Reviewer 2 Report
The study represents a fine contribution for the health evaluation of common eiders.
Findings seemed larger than pathological descriptions, as a diversity of etiologies were detected; aiming to reinforcing the importance of this well-written study, I would suggest a tentative emphasis indicating the broad infectious status, such as "health". Possibly "Health status of Western Baltic Sea bycaught common eiders". But I could not come with a good recommendation.
The common eiders are with declining populations, although in good or moderate nutritional status were most infected with intestinal parasites, and inflammation in the liver, kidneys, intestines and the oesophagus.
I have recommended, although not for the present, for future studies, including these samples, the evaluation of toxic pollutants.
Please refer to the attached copy of the manuscript for details.

Reviewer 3 Report
I am not sure all of the images are fully required such as the pellets but the article is well written and coherent. I see little that needs to be changed other than possibly making it a bit shorter.
